# Causal relationships between obesity and the leading causes of death in women and men

**Jenny C. Censin**[1,2], **Sanne A. E. Peters**[3,4], **Jonas Bovijn**[1,2], **Teresa Ferreira**[1], **Sara L. Pulit**[1,5,6], **Reedik Mägi**[7], **Anubha Mahajan**[2,8], **Michael V. Holmes**[9,10,11☯], **Cecilia M. Lindgren**[1,2,6☯] *

**1** Big Data Institute at the Li Ka Shing Centre for Health Information and Discovery, University of Oxford, Oxford, United Kingdom, **2** Wellcome Centre for Human Genetics, Nuffield Department of Medicine, University of Oxford, Oxford, United Kingdom, **3** The George Institute for Global Health, University of Oxford, Oxford, United Kingdom, **4** Julius Center for Health Sciences and Primary Care, University Medical Center Utrecht, Utrecht, The Netherlands, **5** Department of Genetics, Center for Molecular Medicine, University Medical Center Utrecht, Utrecht, The Netherlands, **6** Program in Medical and Population Genetics, Broad Institute, Cambridge, Massachusetts, United States of America, **7** Estonian Genome Center, Institute of Genomics, University of Tartu, Tartu, Estonia, **8** Oxford Centre for Diabetes, Endocrinology and Metabolism, Radcliffe Department of Medicine, University of Oxford, Oxford, United Kingdom, **9** NIHR Oxford Biomedical Research Centre, Oxford University Hospitals NHS Foundation Trust, John Radcliffe Hospital, Oxford, United Kingdom, **10** Medical Research Council Population Health Research Unit at the University of Oxford, Nuffield Department of Population Health, University of Oxford, Oxford, United Kingdom, **11** Clinical Trial Service Unit & Epidemiological Studies Unit (CTSU), Nuffield Department of Population Health, Big Data Institute Building, Roosevelt Drive, University of Oxford, Oxford, United Kingdom

☯ These authors contributed equally to this work.
* lindgrenpa@bdi.ox.ac.uk

**Data Availability Statement:** Summary-level data for fasting glucose and fasting insulin can be downloaded from https://www.magicinvestigators.org/downloads/. Individual-level data from UK

## Abstract

Obesity traits are causally implicated with risk of cardiometabolic diseases. It remains unclear whether there are similar causal effects of obesity traits on other non-communicable diseases. Also, it is largely unexplored whether there are any sex-specific differences in the causal effects of obesity traits on cardiometabolic diseases and other leading causes of death. We constructed sex-specific genetic risk scores (GRS) for three obesity traits; body mass index (BMI), waist-hip ratio (WHR), and WHR adjusted for BMI, including 565, 324, and 337 genetic variants, respectively. These GRSs were then used as instrumental variables to assess associations between the obesity traits and leading causes of mortality in the UK Biobank using Mendelian randomization. We also investigated associations with potential mediators, including smoking, glycemic and blood pressure traits. Sex-differences were subsequently assessed by Cochran's Q-test ($P_{het}$). A Mendelian randomization analysis of 228,466 women and 195,041 men showed that obesity causes coronary artery disease, stroke (particularly ischemic), chronic obstructive pulmonary disease, lung cancer, type 2 and 1 diabetes mellitus, non-alcoholic fatty liver disease, chronic liver disease, and acute and chronic renal failure. Higher BMI led to higher risk of type 2 diabetes in women than in men ($P_{het} = 1.4 \times 10^{-5}$). Waist-hip-ratio led to a higher risk of chronic obstructive pulmonary disease ($P_{het} = 3.7 \times 10^{-6}$) and higher risk of chronic renal failure ($P_{het} = 1.0 \times 10^{-4}$) in men than women. Obesity traits have an etiological role in the majority of the leading global causes of death. Sex differences exist in the effects of obesity traits on risk of type 2

Biobank cannot be shared publicly because of confidentiality but is available from the UK Biobank (https://www.ukbiobank.ac.uk/) for researches who meet the criteria for access to confidential data. The diabetes case and control definition scripts were kindly provided by the authors to 'Algorithms for the Capture and Adjudication of Prevalent and Incident Diabetes in UK Biobank' (pone.0162388). The other code and GRSs related to this project will be available at https://github.com/lindgrengroup/causal.relationships.between.obesity.and.leading.causes.of.death.in.men.and.women. All other relevant data are within the manuscript and its supporting information files.

**Funding:** JCC is funded by an NDM Prize Studentship (17/18_MSD_1108275) from the Oxford Medical Research Council Doctoral Training Partnership (Oxford MRC DTP; https://www.medsci.ox.ac.uk) and the Nuffield Department of Clinical Medicine (https://www.ndm.ox.ac.uk/), University of Oxford. SAEP is supported by a UK Medical Research Council Skills Development Fellowship (MR/P014550/1). JB is supported by funding from the Rhodes Trust (https://www.rhodeshouse.ox.ac.uk/), Clarendon Fund (http://www.ox.ac.uk/clarendon/about) and the Medical Sciences Doctoral Training Centre (https://www.medsci.ox.ac.uk/), University of Oxford. TF is supported by the NIHR Biomedical Research Centre, Oxford. SLP was funded by a Veni Fellowship (016.186.071; ZonMW; https://www.nwo.nl/) from the Dutch Organization for Scientific Research, Nederlandse Organisatie voor Wetenschappelijk Onderzoek (NWO) during the course of the study. MVH works in a unit that receives funding from the Medical Research Council (MRC; https://mrc.ukri.org/) and is supported by a British Heart Foundation Intermediate Clinical Research Fellowship (FS/18/23/33512; https://www.bhf.org.uk/) and the National Institute for Health Research Oxford Biomedical Research Centre (https://oxfordbrc.nihr.ac.uk). CML is supported by the Li Ka Shing Foundation (https://www.lksf.org/), WT-SSI/John Fell funds (https://researchsupport.admin.ox.ac.uk/), the National Institute for Health Research Biomedical Research Centre, Oxford (https://oxfordbrc.nihr.ac.uk/), Widenlife (H2020-TWINN-2015-692065; https://cordis.europa.eu/), and National Institute of Health (NIH; 5P50HD028138-27; https://www.nih.gov/). Computation used the Oxford Biomedical Research Computing (BMRC) facility, a joint development between the Wellcome Centre for Human Genetics and the Big Data Institute supported by Health Data Research UK and the NIHR Oxford Biomedical Research Centre, and with financial support provided by the

diabetes, chronic obstructive pulmonary disease, and renal failure, which may have downstream implications for public health.

## Author summary

Obesity is increasing globally and has been linked to major causes of death, such as diabetes and heart disease. Still, the causal effects of obesity on other leading causes of death is relatively unexplored. It is also unclear if any such effects differ between men and women. Mendelian randomization is a method that explores causal relationships between traits using genetic data. Using Mendelian randomization, we investigated the effects of obesity traits on leading causes of death and assessed if any such effects differ between men and women. We found that obesity increases the risks of heart disease, stroke, chronic obstructive pulmonary disease, lung cancer, diabetes, kidney disease, non-alcoholic fatty liver disease and chronic liver disease. Higher body mass index led to a higher risk of type 2 diabetes in women than in men, whereas a higher waist-hip ratio increased risks of chronic obstructive pulmonary disease and chronic kidney disease more in men than in women. In summary, obesity traits are causally involved in the majority of the leading causes of death, and some obesity traits affect disease risk differently in men and women. This has potential implications for public health strategies and indicates that sex-specific preventative measures may be needed.

## Introduction

It is increasingly evident that obesity negatively impacts human health and the prevalence of obesity is increasing world-wide [1]. Both overall obesity (body mass index (BMI) >30 kg/m$^2$) and fat distribution (waist-hip-ratio (WHR) >1.0 in men and >0.85 in women indicative of abdominal fat accumulation) have been linked to cardiometabolic diseases and death in observational studies [2–5]. Previous studies have found causal relationships between higher BMI and WHR adjusted for BMI (WHRadjBMI) and type 2 diabetes (T2D) and coronary artery disease (CAD), using a limited number of previously known obesity-associated single nucleotide polymorphisms (SNPs) [6–11]. However, sex-specific relationships are largely unexplored as is the the role that obesity traits play in the leading causes of death beyond these cardiometabolic diseases.

Obesity traits are known to differ between women and men; regional obesity prevalence rates often vary between the sexes [12,13], women have higher SNP-based heritability for WHR [14], and >90% of WHRadjBMI-associated SNPs that show evidence of sexual dimorphism have larger effect sizes in women than men [14]. It has been suggested that fat distribution related traits might be more strongly associated with cardiometabolic outcomes in women, although many previous studies are inconclusive [15–19]. Only a few studies have investigated sex differences in the effect of genetic risk for obesity-related traits on disease risk [6,9,11] and have mostly been restricted to waist-related traits and T2D and CAD, using a limited number of analyses and/or SNPs, but without finding evidence of differences in disease risk between men and women [6,9,11].

Expanding to a larger set of robustly associated SNPs may identify previously undetected sexual heterogeneity in obesity-related disease risk. A sex difference in the effect of obesity traits on major causes of death could signify that the disease burden arising from obesity may

Wellcome Trust Core Award Grant Number 203141/Z/16/Z. The funders had not role in the study design, data collection and analysis, decision to publish, or preparation of the manuscript.

**Competing interests:** SLP has, since the writing of the article, started working for Vertex. MVH has collaborated with Boehringer Ingelheim in research, and in accordance with the policy of the Clinical Trial Service Unit and Epidemiological Studies Unit (University of Oxford), did not accept any personal payment. CML has collaborated with Novo Nordisk and Bayer in research, and in accordance with the policy of University of Oxford, did not accept any personal payment.

be differential in women and men, allowing prioritizing of public health resources and potentially, sex-specific preventative strategies. We therefore investigated the extent to which obesity traits causally impact the risk of the major global causes of death, and whether relationships with disease are differential between women and men, exploiting recent advances in discovery of obesity-associated SNPs [14].

## Methods

### Overview of methods

Sex-specific genetic risk scores (GRSs) were constructed and evaluated for BMI, WHR, and WHRadjBMI using genome-wide significant ($P<5\times10^{-9}$) SNPs from a recent genome-wide association study (GWAS) in the Genetic Investigation of ANthropometric Traits (GIANT) [20,21] and UK Biobank [14,22]. These were then investigated for associations with diseases and risk factors in the UK Biobank using regression in a sex-stratified manner. Obesity GRS-outcome combinations that surpassed the multiple testing correction threshold were then analysed with Mendelian randomization (MR) to compute formal causal estimates, and sexual heterogeneity was assessed. In addition, we performed 2-sample MR for outcomes for which we only had access to summary-level data. We also performed sensitivity analyses to explore the robustness of our findings. For an overview of the methods see Fig 1.

### The UK Biobank

The UK Biobank is a prospective UK-based cohort study, with 488,377 genotyped individuals aged 40–69 when recruited [22]. UK Biobank has a Research Tissue Bank approval (Research

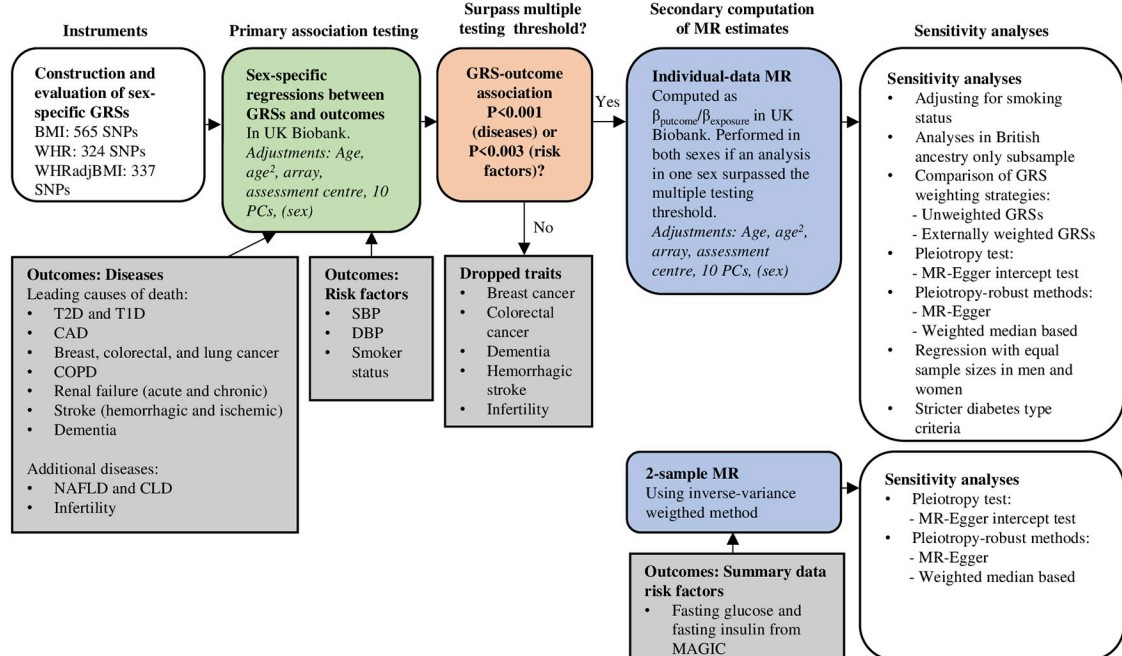

**Fig 1. Overview of methods.** BMI, body mass index; CLD, chronic liver disease; COPD, chronic obstructive pulmonary disease; DBP, diastolic blood pressure; GRS, genetic risk score; MAGIC, Meta-Analyses of Glucose and Insulin-related traits Consortium; MR, Mendelian randomization; NAFLD, non-alcoholic fatty liver disease; PC, principal component; SBP, systolic blood pressure; T1D, type 1 diabetes; T2D, type 2 diabetes; WHR, waist-hip ratio; WHRadjBMI, waist-hip ratio adjusted for body mass index.

Ethics Committee reference 16/NW/0274, this study's application ID 11867), and all participants gave informed consent.

Genotyping, primary genotype quality control, and imputation were performed by the UK Biobank, described in detail elsewhere [23]. Briefly, the UK Biobank samples were genotyped using one of two different arrays, the UK biobank BiLEVE array or the UK biobank array [23]. The genotype data were then imputed using the HRC reference panel [24] or a merged reference panel consisting of UK10K [25] and 1000 Genomes phase 3 [26], with preference given to the HRC reference panel [23]. The imputed genotype data in BGEN v1.2 format were converted into hard calls (--hard-call-threshold 0.1) using PLINK v2.00aLM [14,27,28]. We then performed post-imputation quality control [14,27]. Only SNPs with imputation info score >0.3, minor allele frequency ≥0.01%, and Hardy-Weinberg equilibrium exact test threshold $P \geq 1 \times 10^{-6}$ were kept. In addition, SNPs with a missing call rate >0.05 and non-biallelic SNPs were excluded (for details see S1 Text).

General sample quality control was performed (for details see S1 Text) and samples of non-European ancestry excluded, resulting in a final sample size of up to 423,507 individuals. Participant characteristics are in Table A in S1 Text.

## Instruments

We evaluated several approaches to construct sex-specific GRSs for BMI, WHR, and WHRadjBMI (Fig A-B in S1 Text). The approach with the highest ranges of trait variance explained and F-statistics for the relevant obesity trait, and with no demonstrable heterogeneity between men and women, was selected as the main model. In this model, GRSs were constructed by combining the primary ("index") genome-wide significant ($P < 5 \times 10^{-9}$) SNPs in the men, women, or combined-sexes analyses in the largest GWAS available with sex-specific European summary statistics, a meta-analysis of GIANT [20,21] and the UK Biobank (Fig 2, S1 Text and S1 Table) [14,22]. Primary SNPs were identified in the original GWAS [14] by proximal and joint conditional analysis using GCTA in associated loci. Associated loci were established around top SNPs associated with the obesity trait $P < 5 \times 10^{-9}$, and included all SNPs associated with the obesity trait $P < 0.05$, within ±5 Mb of the top SNPs, and in linkage disequlibrium (LD; $r^2 > 0.05$) with the top SNP; overlapping loci were merged [14].

We then kept the SNP with the lowest combined-sexes P-value within each 1 Mb sliding window to limit correlation between SNPs discovered in different sex-strata in each obesity trait. We excluded non-biallelic SNPs (N = 2), SNPs that failed quality control (N = 3), and one SNP per pair with long-distance linkage disequilibrium ($r^2 > 0.05$, N = 2) (S1 Text). For the combined-sexes analyses, SNPs were weighted using estimates from the combined-sexes European meta-analyzed GWASs. For the men- and women-only analyses, SNPs were weighted by their sex-specific European estimates. All SNPs were orientated so that the effect allele corresponded to a higher level of the investigated obesity trait. Genetic risk scores were then computed using PLINK v1.90b3 (--score, with sum option) [28].

## Exposures: Obesity traits

Baseline measurements were used for BMI, WHR, and WHRadjBMI. They were then standardized by rank inverse normal transformation of the residuals after regression of the trait on baseline age, $age^2$, assessment centre, and, if applicable, sex. This was done separately in the men and women only analyses, but jointly in the combined analyses, after any sample quality exclusions (S1 Text). For WHRadjBMI we also adjusted for BMI.

**Fig 2. SNP- and weight selection flowchart with number of SNPs for each obesity trait.** SNPs were selected by combining the primary ("index") variants associated with the obesity traits $P<5\times10^{-9}$ in a meta-analysis of GIANT and UK Biobank [14,22]. All SNPs were weighted by their sex-specific European estimates for the men- and women-specific analyses, and by the combined-sexes European estimates for the combined-sexes analyses, using estimates from the original genome-wide association study. BMI, body mass index; SNP, single nucleotide polymorphism; WHR, waist-hip-ratio; WHRadjBMI, waist-hip-ratio adjusted for body mass index.

## Outcomes: Diseases

We investigated associations between the three obesity traits (BMI, WHR, and WHRadjBMI) with all non-communicable diseases on the World Health Organization's (WHO) list of leading mortality causes world-wide and in high-income countries [29]; CAD, stroke (including ischemic, hemorrhagic, and of any cause), chronic obstructive pulmonary disease (COPD), dementia, lung cancer, T2D and type 1 diabetes (T1D), colorectal cancer, renal failure

(including acute, chronic and of any cause) and breast cancer in women (Table B in S1 Text). In addition, we included infertility, non-alcoholic fatty liver disease (NAFLD) and chronic liver disease (CLD) as they have previously been linked to obesity and represent important and increasing burdens of disease [30–36]. For T2D and T1D, we drew case definitions from a validated algorithm for prevalent T2D and T1D (using "probable" and "possible" cases) and those the algorithm denoted as "diabetes unlikely" were used as controls [37]. For CAD, we used the same case and control definitions as a large GWAS [38]. Case and control criteria for the other disease outcomes were defined using self-reported data, data from an interview with a trained nurse, and hospital health outcome codes (also including death and cancer registry) in discussion between two licensed medical practitioners (Table B in S1 Text). For CAD, acute renal failure, chronic renal failure, stroke of any cause, ischemic stroke and hemorrhagic stroke, exclusions for certain codes were also made in the control groups after defining the case groups. All available information was used to decide on case and control status (except for diabetes), including information collected after the baseline assessment (including repeat assessments and hospital health outcome codes).

## Outcomes: Risk factors

To assess the relationship of obesity traits with risk factors that might mediate the disease associations, we also investigated associations between the obesity traits and the cardiometabolic risk factors systolic blood pressure (SBP), diastolic blood pressure (DBP), fasting glucose (FG), fasting insulin (FI), and smoking status.

The mean of the baseline measurements was used for SBP and DBP. Fifteen mmHg to SBP and 10 mmHg to DBP were added if blood pressure lowering medications were used (defined as self-reported use of such in data-fields 6153 and 6177), as in previous blood pressure GWASs and as suggested in simulation studies [39,40]. We then standardized the blood pressure traits by rank inverse normal transformation of the residuals after regression of the trait on baseline age, $age^2$, assessment centre, and, if applicable, sex. This was done separately in the men and women only analyses, but jointly in the combined analyses, after any sample quality exclusions (S1 Text).

Smoking status was defined as self-report of being a current or previous smoker or having smoked or currently smoking (most days or occasionally; any code 1 or 2 in any of the data fields 1239, 1249, and 20116 for the baseline assessment).

Sex-specific summary-level data for plasma FG (in mmol/L, untransformed, corrected to plasma levels using a correction factor of 1.13 if measured in whole blood in the original GWAS) and serum FI (in pmol/L, ln-transformed) were kindly provided by the Meta-Analyses of Glucose and Insulin-related traits Consortium (MAGIC) investigators and can be downloaded from https://www.magicinvestigators.org/downloads/ [41]. SNPs in chromosome:position format were converted to rsIDs using the file All_20150605.vcf.gz from the National Center for Biotechnology Information (NCBI) [42] (available at ftp://ftp.ncbi.nih.gov/snp/organisms/archive/human_9606_b144_GRCh37p13/VCF/). All SNPs were then updated to dbSNP build 151 using the file RsMergeArch.bcp.gz, also from the NCBI [42] (available at ftp://ftp.ncbi.nlm.nih.gov/snp/organisms/human_9606/database/organism_data/).

## Statistical analyses: Evaluation of instruments

The GRSs were first assessed if they were robustly associated with their respective obesity traits by computing trait variance explained and the F-statistics using linear regression (Table C in S1 Text). Adjustments were made for array type and 10 PCs, as we had previously adjusted for age, $age^2$, assessment centre, and if applicable sex and BMI, in the rank inverse normal

transformation of the obesity traits. Sexual heterogeneity was assessed using Cochran's Q test [43], with the $P_{het}$-threshold set at <0.002 (= 0.05/21) for 21 male-female instrument comparisons.

## Statistical analyses: Primary association testing

We then explored the associations of the sex-specific GRSs with diseases and risk factors available in the UK Biobank to see if there were any indications of a causal relationship between the obesity traits and the outcomes [44]. Logistic regression was used for disease outcomes and smoking status and linear regression was used for SBP and DBP. Adjustments were made for baseline age, $age^2$, array type, assessment centre, 10 principal components, and sex if applicable, for all traits when in clinical units and for binary outcomes, and array and 10 principal components if rank inverse normal transformed (where adjustment for age, $age^2$, assessment centre, and if applicable sex had already been performed in the rank inverse normal transformation of the residuals).

Associations between the sex-specific GRSs with outcome traits that surpassed our P-value thresholds were taken forward for MR to more formally quantify the effect of the obesity trait on the outcome. For the obesity trait-disease analyses, the P-value threshold was set at <0.001 (= 0.05/51) for 51 obesity trait-disease combinations investigated in the study. For the obesity trait-risk factor analyses, the P-value threshold was set at <0.003 (= 0.05/15), for the total of 15 obesity trait-risk factor combinations investigated in the study (as we also assessed fasting glucose and fasting insulin using summary data). If a combined-sexes regression analysis identified evidence against the null hypothesis it was taken forward for MR; if a regression analysis identified evidence against the null hypothesis in either men or women, MR was performed in both the men and women-only stratified analyses so sexual heterogeneity could be assessed. Since we conducted the MR analyses both with and without adjusting for smoking status, we conducted MRs for all obesity traits with smoking status for completeness.

## Statistical analyses: Secondary computation of MR estimates

Individual-level MR was performed using the Wald method, with the instrumental variable estimate being the ratio between the computed betas for the outcome and risk factor regressed separately on each GRS [45]. In this step, logistic regression was used for binary outcomes using the log(odds ratio(OR)) in the ratio and linear regression used for continuous outcomes. For the binary outcomes, MR regressions of the obesity traits on the GRSs were performed only including the controls for each outcome. Standard errors were adjusted to take the uncertainty in both regressions into account by using the first two terms of the delta method [44,46,47].

Adjustments were made for baseline age, $age^2$, array type, assessment centre, 10 principal components, and sex if applicable, for all traits when in clinical units and for binary outcomes, and array and 10 principal components if rank inverse normal transformed (where adjustment for age, $age^2$, assessment centre, and if applicable sex had already been performed in the rank inverse normal transformation of the residuals).

For the obesity trait-disease analyses, the P-value threshold was set at <0.001 (= 0.05/51) for the total of 51 obesity trait-disease combinations investigated in the study. For the obesity trait-risk factor analyses, the P-value threshold was set at <0.003 (= 0.05/15), for the total of 15 obesity trait-risk factor combinations investigated in the study.

## Statistical analyses: 2-sample Mendelian randomization

We performed 2-sample summary-level MRs for the potential risk factors FG and FI directly, as we only had summary-level data for these traits. The main estimates were computed using

the inverse-variance weighted (IVW) method. We then computed MR-Egger and weighted median estimates as a sensitivity analysis [48–51]. The P-value threshold was set at <0.003 (= 0.05/15) for the total of 15 obesity trait-risk factor combinations investigated in the study (including individual-level MRs for DBP, SBP, and smoking status).

### Statistical analyses: Assessment of sexual heterogeneity

Sexual heterogeneity between male and female estimates from the regressions and the MRs was assessed using P-values from Cochran's Q test [43]. To facilitate comparisons between the obesity traits and sex-strata, MR estimates were computed per 1 standard deviation (SD) higher obesity trait. The $P_{het}$-threshold for obesity trait-disease analyses was set at <0.001 (= 0.05/48) for 48 male-female estimates comparisons, since breast cancer was investigated in women only. For the obesity trait-risk factor analyses, the $P_{het}$-threshold was set at <0.003 (= 0.05/15) for the 15 male-female estimates comparisons.

### Sensitivity analyses

We performed several sensitivity analyses to ascertain robustness; we conducted (a) analyses adjusting for smoking status and (b) analyses restricted to those of genetically confirmed British ancestry only (S1 Text). We also (c) evaluated the robustness of the MR findings by comparing different weighting strategies, including use of unweighted and externally weighted (using weights from the GIANT 2015 studies [20,21]) GRSs, and (d) investigated for pleiotropy and performed more pleiotropy-robust sensitivity analyses [50,51] (S1 Text). We also (e) performed logistic regressions using the same number of cases and controls in men and women for the disease outcomes and (f) conducted analyses using stricter T2D and T1D case definitions (S1 Text). In addition, we (g) recomputed the MR estimates for the obesity trait-disease combinations with evidence of sexual heterogeneity using additional SNP-selection and weighting approaches (S1 Text).

### Software

The diabetes case and control definition scripts were kindly provided by the authors to 'Algorithms for the Capture and Adjudication of Prevalent and Incident Diabetes in UK Biobank' [37]. The other code and GRSs related to this project will be available at https://github.com/lindgrengroup/causal.relationships.between.obesity.and.leading.causes.of.death.in.men.and.women. The genotype data was handled using PLINK v2.00aLM and PLINK v1.90b3 [28] (S1 Text). Further data handling was performed in Python 3.5.2 [52] using the packages "pandas" [53] and "numpy" [54], R version 3.4.3 [55] and the package "dplyr" [56], bash version 4.1.2(2) [57] and awk [58]. Statistical analyses and plots were performed using R version 3.4.3 [55] and packages "ggplot2" [59], "mada" [60], "dplyr" [56], "gridExtra" [61], "lattice" [62], "grid" [55], "grDevices" [55], "ggpubr" [63], and "MendelianRandomization" [48].

## Results

### Evaluation of genetic risk scores

The three GRSs included 565 SNPs for BMI, 324 for WHR and 337 for WHRadjBMI. Trait variance explained varied between 2.5–7.1% and the F-statistic between 4,921–26,466, depending on trait and sex-stratum (Table C in S1 Text).

## Primary association testing: Disease outcomes and risk factors

We first assessed the associations of the GRSs with diseases and risk factors using linear or logistic regression (Table D-F and Fig C in S1 Text). Obesity GRSs were associated with diabetes, CAD, COPD, lung cancer, stroke (of any cause and ischemic), renal failure, liver diseases, as well as blood pressure traits (Table D-F and Fig C in S1 Text). Several instruments also associated with smoking status and with higher estimate magnitudes in men than in women for both BMI and WHR (BMI: $P_{het}$ = 2.2×10$^{-4}$; WHR: $P_{het}$ = 8.0×10$^{-14}$; WHRadjBMI $P_{het}$ = 0.008) (Table E in S1 Text). The GRS-outcome (diseases and risk factors) associations that surpassed correction for multiple testing were taken forward for MR to compute formal estimates of effect–if surpassed in one sex only, both men and women-only stratified analyses were conducted so that sexual heterogeneity could be assessed.

No obesity GRS showed evidence for association in any sex-strata for colorectal cancer, breast cancer (investigated in women only), dementia, hemorrhagic stroke, and infertility, and these endpoints were therefore not taken forward for computation of formal MR estimates (Table F and Fig C in S1 Text).

## Mendelian randomization of obesity with disease outcomes: All individuals

Obesity traits were causally implicated with diseases that represent the major causes of death (Figs 3 and 4). All measures of obesity were strongly causally related to risk of CAD OR ranging from 1.39 for WHRadjBMI to 1.74 for WHR in the combined analyses per 1-SD higher obesity trait). For stroke, both BMI and WHR conferred higher risk (ORs 1.40 and 1.34, respectively). Strong effects were seen for all obesity traits with T2D (OR range 2.10 to 3.62) and BMI also associated with risk of T1D (OR 1.68), with highly similar results using stricter diabetes type criteria (S1 Text). Obesity traits increased the risk of kidney disease, including both acute (ORs 1.55 for WHR and 1.81 for BMI) and chronic (ORs 1.72 for WHR and 1.81 for BMI) renal failure. Strong effects were also seen for risk of NAFLD (OR range 1.60–2.89) and CLD (ORs 1.64 for BMI and 1.85 for WHR).

Measures of obesity also causally impacted on risks of COPD (OR 1.66 for BMI and 1.45 for WHR) and lung cancer (BMI OR 1.33). As several GRSs had associated with smoking status, we repeated the individual-level MRs adjusting for smoking status to assess potential mediation. Whereas most obesity-disease associations were largely similar, adjusting for smoking status resulted in diminished magnitudes of effect for COPD and lung cancer, suggesting potential mediation (Table G in S1 Text).

Sensitivity analyses, including restricting to those of genetically confirmed British ancestry only, use of different weighting strategies, analyses using more pleiotropy-robust methods, using the same number of cases and controls in men and women, and use of more stringent diabetes case definitions supported the main findings (Tables H,I and Fig D-F in S1 Text).

## Mendelian randomization of obesity with disease outcomes: Sex-stratified analyses

Five out of the 24 obesity trait-disease associations differed between women and men (Fig 3). The effect of BMI on T2D risk was higher in women than men, with strong evidence for sexual heterogeneity (women: OR 3.77; 95% CI 3.38–4.20, P = 1.7×10$^{-128}$; men: OR 2.79; 95% CI 2.58–3.03, P = 5.7×10$^{-135}$, per 1-SD higher BMI, $P_{het}$ = 1.4×10$^{-5}$). This sexual heterogeneity could also be observed in sensitivity analyses where the number of cases in women and men was similar ($P_{het}$ = 5.6×10$^{-6}$) (Table I in S1 Text) and when performing the analysis using stricter T2D diagnosis criteria ($P_{het}$ = 3.5×10$^{-5}$) (S1 Text).

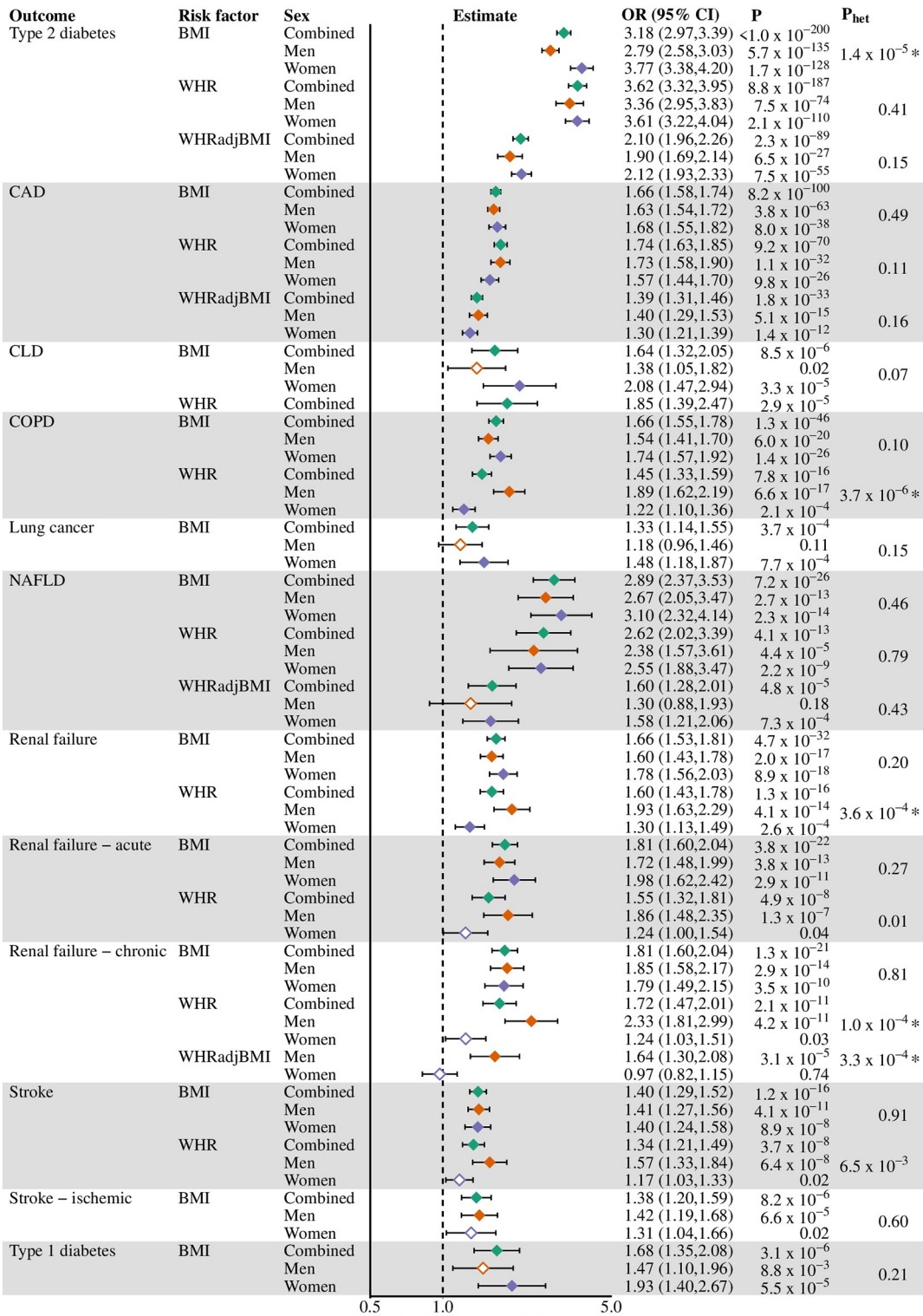

**Fig 3. Causal effects of obesity traits on disease outcomes, overall and stratified by sex.** Endpoints that showed an association with obesity GRSs were taken forward for Mendelian randomization, with estimates reported as odds ratio (95% CI) per 1-SD higher obesity trait. Filled diamonds indicate that the P-value for the obesity trait to disease endpoint surpasses our threshold for multiple testing; empty diamonds indicate that the P-value does not surpass this threshold (Bonferroni-adjusted P-value-threshold set at <0.001 (= 0.05/51) for 51 obesity trait-disease outcome combinations in the study). * denotes

that the P-value for heterogeneity (from Cochran's Q test) surpasses our threshold for multiple testing; $P_{het}$-threshold set at <0.001 (= 0.05/48) for 48 male-female comparisons in the study (fewer since breast cancer analyses were performed in women only). Green diamond, combined-sexes estimates; orange diamond, male estimates; purple diamond, female estimates; BMI, body mass index; CAD, coronary artery disease; COPD, chronic obstructive pulmonary disease; NAFLD, non-alcoholic fatty liver disease; SD standard deviation; WHR, waist-hip-ratio; WHRadjBMI, waist-hip-ratio adjusted for body mass index.

The estimate for the effect of higher WHR on COPD risk was greater in men than in women (men: OR 1.89; 95% CI 1.62–2.19, P = $6.6 \times 10^{-17}$; and women: OR 1.22; 95% CI 1.10–1.36, P = $2.1 \times 10^{-4}$, per 1-SD higher BMI, $P_{het}$ = $3.7 \times 10^{-6}$). Similarly, higher WHR was also associated with a greater risk of being a smoker in men than in women ($P_{het}$ = $2.2 \times 10^{-14}$) (Table J in S1 Text). Despite this, the effect estimates for the effect of WHR on COPD risk remained higher in men after adjusting for smoking ($P_{het}$ = $8.8 \times 10^{-5}$) (Table G in S1 Text).

There was also evidence of higher WHR increasing the risk of renal failure more in men than in women ($P_{het}$ = $3.6 \times 10^{-4}$). This sexual heterogeneity may originate from a risk difference in the effect of WHR on chronic renal failure, as men had higher risk estimates than

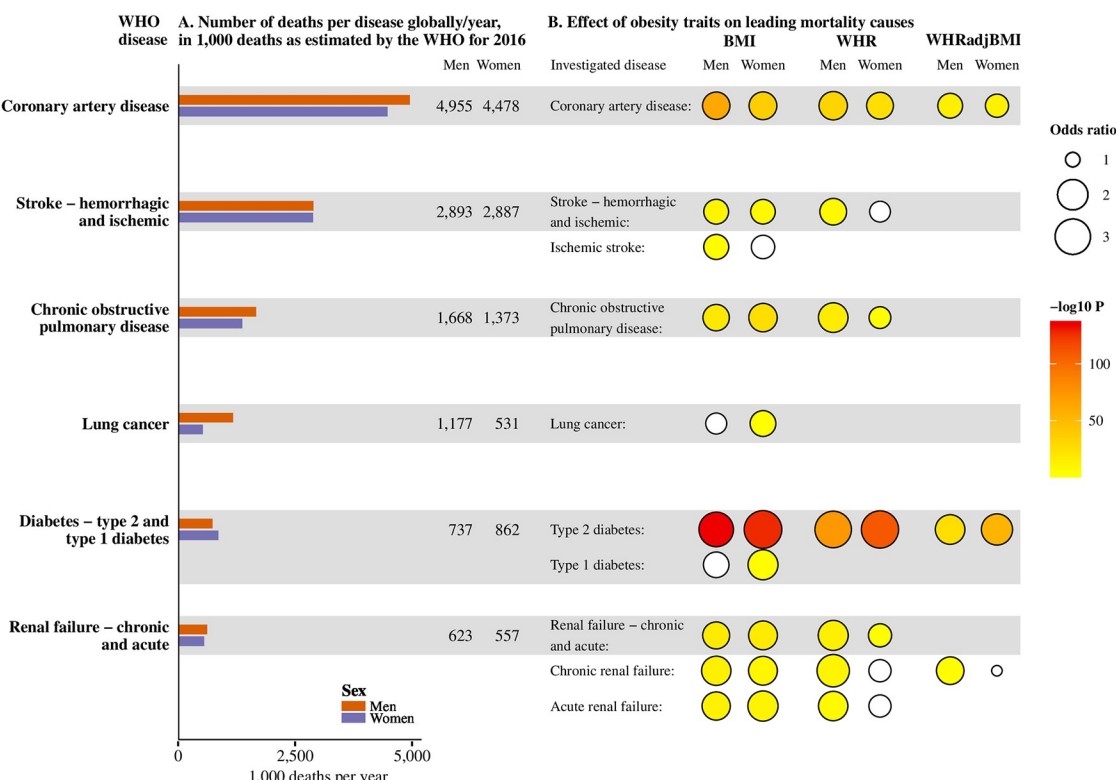

**Fig 4. Overview of the sex-specific effect magnitudes and strengths of association of obesity traits on leading causes of death.** Leading causes of death defined as non-communicable diseases on the WHO top 10 lists of causes of death, globally and in high-income countries, with additional separate analyses for subclasses of stroke, diabetes, and renal disease. No obesity trait (BMI, WHR, or WHRadjBMI) genetic risk score associated with dementia, colorectal cancer, breast cancer (investigated in women only) or hemorrhagic stroke–these are not shown on the plot (for the regression results see Table F in S1 Text). (A) Total number of deaths globally, in 1,000 deaths, as estimated by the WHO for 2016 [64], stratified by sex. For diabetes, estimates for annual number of deaths are for type 1 and type 2 diabetes combined. (B) Obesity trait-disease combinations taken forward for Mendelian randomization showed with circles. Mendelian randomization associations with P-values surpassing our threshold in yellow to red fill depending on P-value (-log10 P-value), white fill indicates a P-value not surpassing our threshold. The size of the circles corresponds to the magnitude of the odds ratio estimate for the Mendelian randomization estimate. Estimates and P-values from the MR analyses of the obesity traits with the disease outcomes using the sex-specific estimates approach. BMI, body mass index; P, P-value; WHR, waist-hip-ratio; WHRadjBMI, waist-hip-ratio adjusted for body mass index; WHO, World Health Organization.

women in that analysis ($P_{het}$ = $1.0 \times 10^{-4}$, with a similar sexual heterogeneity seen in the effect of WHRadjBMI). A 1-SD higher WHR was also associated with higher magnitudes of risk for acute renal failure in men than women, although the $P_{het}$-value did not pass our $P_{het}$-threshold (men: OR 1.86; 95% CI 1.48–2.35, P = $1.3 \times 10^{-7}$; women: OR 1.24; 95% CI 1.00–1.54, P = 0.04, $P_{het}$ = 0.01).

Sensitivity analyses using different GRS weighting strategies strongly supported sex-differences in the effect of BMI on T2D and WHR on chronic renal failure and COPD, but only weakly supported a sex-difference in the effect of WHR on renal failure of any cause (Fig D,E and G-I in S1 Text).

## Potential mechanisms

To identify associations of obesity markers with risk factors that could mediate the disease risks, we assessed the relationship of obesity traits with blood pressure (SBP, DBP), glycemic traits (FG, FI), and smoking status (Figs 5 and 6, Tables J-N in S1 Text). All obesity traits causally increased SBP, DBP, FG and FI (Fig 5). The increase in DBP arising from elevated BMI was greater in women than men ($P_{het}$ = $5.2 \times 10^{-5}$).

We computed MR estimates for smoking status in all sex-strata for completeness, since we performed MRs adjusted for smoking status as a sensitivity analysis. BMI and WHR both associated with higher risk of being a smoker, with the effect magnitudes being larger in men than women (BMI $P_{het}$ = $8.4 \times 10^{-4}$; WHR $P_{het}$ = $2.2 \times 10^{-14}$) (Fig 6). WHRadjBMI was only associated with smoking status in men.

## Discussion

Our work shows that obesity is causally implicated in the etiology of two thirds of the globally leading causes of death from non-communicable diseases [29]. Furthermore, we identify that for some diseases, obesity conveys altered magnitudes of risk in men and women. Such sexual dimorphism could be observed in the effects of BMI on T2D and waist-related traits on COPD and renal failure. These findings have potential implications for public health policy.

### Diabetes

Obesity traits were causally related to higher risk of T2D, in keeping with previous studies [6–11,19,65]. We could not detect a sex difference in risk of T2D from higher WHR or WHRadjBMI. Even though some observational studies have suggested that WHR may be a stronger predictor of T2D risk in women than in men [18,19], studies investigating the effect on T2D risk from genetic predisposition to higher WHRadjBMI have not found evidence of sexual heterogeneity [6,9,11]. In contrast, we found that BMI conferred a higher T2D risk in women than in men. Whereas men tend to be diagnosed with T2D at lower BMI than women [66], there may be a stronger association between increase of BMI and T2D risk in women than in men [15,18,65,67–70]. Whether this reflects a stronger causal effect of BMI on T2D risk in women has hitherto been unknown. There was no evidence for sexual heterogeneity of the causal effect of BMI on potential glycemic trait risk mediators (FG and FI). There have been indications of higher BMI being observationally associated with lower insulin sensitivity more in men than in women, but this observed sex-difference may not reflect a causal pathway or we are not capturing it by our glycemic measurements [71–73]. We also found evidence of BMI causally increasing risk of T1D. Previous observational [74] and MR [75] studies have implicated childhood BMI in risk of T1D. As SNPs associated with adult BMI have also been found to affect childhood BMI [75,76], our results may reflect the consequences of childhood

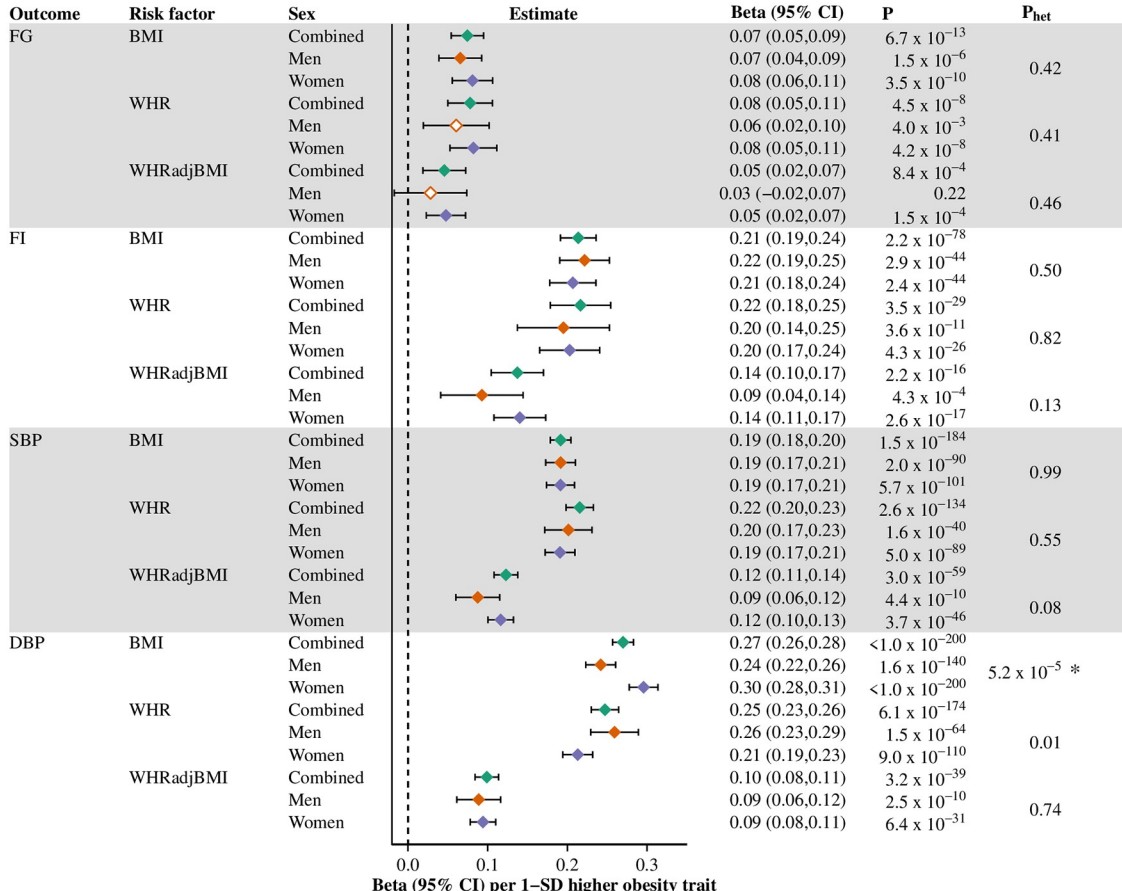

**Fig 5. Causal effects of obesity traits on continuous risk factors, overall and stratified by sex.** The obesity-risk factor combinations brought forward for Mendelian randomization. Estimates in plasma mmol/L levels for FG, serum pmol/L levels (ln-transformed) for FI and SD-units for SBP and DBP, per 1-SD higher obesity trait. Filled diamonds indicate that the P-value for the obesity trait to risk factor endpoint surpasses our threshold for multiple testing; empty diamonds indicate that the P-value does not surpass this threshold (Bonferroni-adjusted P-value-threshold set at <0.003 (= 0.05/15) for 15 obesity trait-risk factor combinations in the study). * denotes that the P-value for heterogeneity (from Cochran's Q test) surpasses our threshold for multiple testing; $P_{het}$-threshold set at <0.003 (= 0.05/15). Green diamond, combined-sexes estimates; orange diamond, male estimates; purple diamond, female estimates; BMI, body mass index; DBP, diastolic blood pressure; FG, fasting glucose; FI, fasting insulin; SBP, systolic blood pressure; WHR, waist-hip-ratio; WHRadjBMI, waist-hip-ratio adjusted for body mass index.

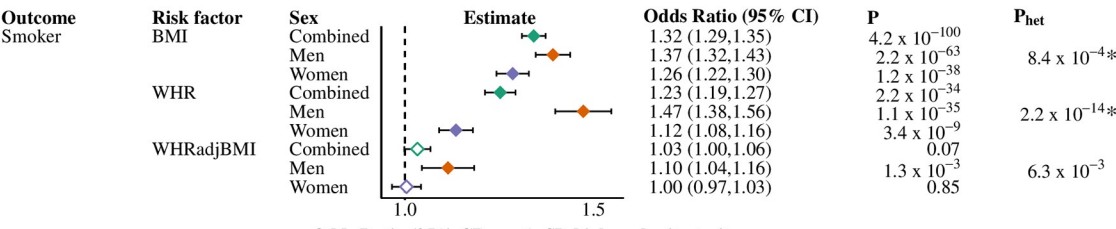

**Fig 6. Causal effects of obesity traits on having been or being a smoker, overall and stratified by sex.** Estimates given in odds ratio (95% CI) per 1-SD higher obesity trait. Filled diamonds indicate that the P-value for the obesity trait to disease endpoint surpasses our threshold for multiple testing; empty diamonds indicate that the P-value does not surpass this threshold (Bonferroni-adjusted P-value-threshold set at <0.003 (= 0.05/15) for 15 obesity trait-risk factor combinations in the study). * denotes that the P-value for heterogeneity (from Cochran's Q test) surpasses our threshold for multiple testing; $P_{het}$-threshold set at <0.003 (= 0.05/15). Green diamond, combined-sexes estimates; orange diamond, male estimates; purple diamond, female estimates; BMI, body mass index; SD standard deviation; WHR, waist-hip-ratio; WHRadjBMI, waist-hip-ratio adjusted for body mass index.

BMI on T1D rather than adult BMI. The results were robust to use of stricter diabetes case definition criteria, minimizing risk of erroneous findings due to misclassification of diabetes type.

## Cardiovascular disease

All obesity traits increased risk of CAD in both sexes, with no difference detected in the magnitude of effect between women and men. The associations with risk of CAD in men and women combined are consistent with previous studies [4,6–8,10,11,15,17]. While observational studies have indicated that waist-related traits may be more strongly associated with cardiovascular disease in women than men, they have not been conclusive [15,17,77,78]. However, a recent study [11] investigated the effect of higher WHRadjBMI, lower gluteofemoral fat distribution, and higher abdominal fat distribution, proxied by genetic variants, on CAD and T2D risk and found no evidence that relationships differed between men and women, similar to our findings. BMI and WHR have previously been observationally associated with risk of stroke [79–81] and a previous MR study found a causal effect of BMI on ischemic stroke [82]. However, some studies have found WHR to be an epidemiological risk factor for stroke in men only [79,80]. Our results confirm BMI as a causal risk factor for stroke (and particulary ischaemic stroke) in both men and women. In women, the effects of WHR were directionally consistent with harm, but the estimates were imprecise, probably reflecting insufficient power in the sex-stratified analysis.

## Lung disease

BMI and WHR increased the risk of COPD and BMI increased the risk of lung cancer; a likely common mechanism is smoking. BMI has previously been implicated in the aetiology of COPD, but is not an established epidemiological risk factor [7,83–85]. Obesity may directly contribute to COPD as its diagnosis is partly based on spirometry values, and obesity is associated with lower lung function [85,86]. Higher BMI also increased risk of lung cancer in our study, similar to a previous MR study [87]. Observational studies tend to identify associations between smoking and lower body weight, but whereas smoking lowers body weight, higher BMI is associated with increased smoking [87–90]. We found associations between particularly BMI and WHR with smoking propensity. To assess mediation, we therefore conducted analyses adjusting for smoking status. This attenuated the associations between the obesity markers and risks of COPD and lung cancer, providing some evidence that smoking may lie on the causal pathway between obesity and lung disease. This diminution does not discredit the validity of the MR analyses unadjusted for smoking provided that the obesity instruments only affect smoking propensity through altered obesity [91]. Rather, they suggest that higher BMI impacts on disease beyond the immediate physiological effects of obesity: by altering human behavior (i.e. increased smoking, possibly motivated as a weight loss strategy [92,93]) and this increased propensity to smoking has additional, far-reaching, deleterious effects on human health, as evidenced by the higher risks of serious lung disease. Higher WHR was associated with greater risks of both COPD and being a smoker in men than in women. Whilst the sex difference in the effect of WHR on COPD persisted after adjustment for smoking status, we cannot rule out that WHR has a higher effect on COPD in men than women through its effect on smoking propensity, but that our smoking phenotype does not fully capture the life-long effects of smoking in men as compared to women.

## Kidney disease

Our results also provide further evidence for a causal role of obesity traits in both acute and chronic renal disease—previous MR studies assessing these relationships have not been

conclusive [6,7,94–96]. Obesity may affect chronic renal disease through a number of mechanisms, including structural changes in the kidney, higher blood pressure and through higher risks of mediating diseases, such as T2D [96–99]. We found central fat distribution (as measured by WHR and WHRadjBMI) to have higher effects on chronic renal failure in men than in women, with evidence of sexual heterogeneity, but the reason for this sex difference is unclear.

## Liver disease

Obesity traits associated with increased risk of NAFLD and CLD (important and emerging causes of chronic disease and mortality [32–35]), with the effect of obesity on CLD possibly mediated by NAFLD [33]. A previous MR study found BMI to increase hepatic triglyceride content [100]. Our study confirms a role of both general obesity and central fat distribution in NAFLD and CLD using an MR design. This strengthens evidence of a causal effect and, given that the prevalence of NAFLD is on the rise (with an estimated one billion people affected globally [101]), emphasizes the potential of an increased burden of CLD as a consequence of global obesity [1,32–35].

## Other diseases

No obesity GRS showed evidence of association in any sex-strata for colorectal cancer, dementia, hemorrhagic stroke, breast cancer (investigated in women only) and infertility. Whereas this may be due to obesity traits not having an effect on these diseases, there are also other potential explanations. For example, previous observational studies have indicated that higher BMI may be protective in premenopausal breast cancer but harmful in postmenopausal breast cancer [102,103]. It is thus possible that opposing effect directions of BMI on breast cancer risk depending on menopausal state counteract each other in our study. For some diseases, it may be that we have too few cases to detect an association. For example, previous MR studies found evidence in support of a causal role of BMI with risk of colorectal cancer but used larger sample sizes [104,105], meaning that our findings may be false negatives due to inadequate statistical power: we also had relatively few cases for dementia, hemorrhagic stroke, and infertility.

## Strengths and limitations

Genetic instruments should only affect the outcome through the risk factor of interest and not through any confounders [106,107]. We performed sensitivity analyses (MR-Egger, weighted-median based methods) more robust to such bias, which supported the main findings [50,51].

If instruments are weakly associated with their respective traits, it can introduce bias in MR studies [108]. We therefore only used instruments strongly associated with their respective risk factor, and performed sensitivity analyses using a variety of SNP-selection and weighting approaches, including unweighted and externally weighted scores, which also supported the main results [44,108,109]. These sensitivity analyses and the strict P-value threshold to denote evidence in support of the presence of sexual heterogeneity should also lessen the risk of the observed sex-differences being due to winner's curse, although such bias cannot be completely ruled out.

Recent studies have also indicated that there may be slight population stratification in both GIANT and UK Biobank, although such bias is likely to be minor [110,111]. Our study was restricted to individuals of Europeans ancestry; limiting our analyses to those of British ancestry only, yielded near-identical results. Associations between the obesity traits and outcomes may differ in other ancestries.

It is possible that our genetic instrument for WHRadjBMI might show features of collider bias whereby SNPs included in the GRS associate with both higher WHR and lower BMI leading to potentially spurious findings [112]. We note that a recent GWAS [14] evaluated the potential for collider bias in the WHRadjBMI GWAS and found limited evidence for such, although the GRS was associated with higher WHR and lower BMI. The directional consistency of associations between WHR and WHRadjBMI and disease endpoints in our analysis suggests that collider bias is unlikely to represent a major source of error in this study.

Most MR studies to date have weighted instruments for the exposure by estimates derived from analyses in which women and men were combined and thus investigated the average causal effect of both sexes. However, this may obscure causal effects that differ between women and men, and can, in addition, cause less precise estimates [109] or an over/underestimation of the effect of the instrument on the exposure in each sex. Our results indicate that weighting SNPs by their sex-specific estimates improves instrument strength and precision compared to using weights derived in a combined-sexes sample. While the work we perform makes important inroads into the development of sex-specific Mendelian randomization approaches, we recognize that, for example, imbalances in the proportion of women and men in included studies, differential availability of summary-level sex-stratified GWAS data and the potential for biases to operate differentially between women and men pose additional complexities in deciphering underlying causal effects.

## Conclusion

Our results implicate obesity in the etiology of the leading causes of death globally, including CAD, stroke, type 2 and 1 diabetes, COPD, lung cancer and renal failure, as well as NAFLD and CLD. This increased risk arising from obesity differs between men and women for T2D, renal failure and COPD. Our findings emphasize the importance of improved preventative measures and treatment of obesity-related disorders and implies that women and men may experience different disease sequelae from obesity, with potential implications for provision of health services and public health policy.

## Supporting information

**S1 Text. Supporting information.**
(DOCX)

**S1 Table. The genetic risk scores used in the main analyses.**
(XLSX)

## Acknowledgments

Computation used the Oxford Biomedical Research Computing (BMRC) facility, a joint development between the Wellcome Centre for Human Genetics and the Big Data Institute supported by Health Data Research UK and the NIHR Oxford Biomedical Research Centre. The views expressed are those of the author(s) and not necessarily those of the NHS, the NIHR or the Department of Health.

We thank the UK Biobank (application 11867; http://www.ukbiobank.ac.uk/).

## Author Contributions

**Conceptualization:** Sanne A. E. Peters, Michael V. Holmes, Cecilia M. Lindgren.

**Data curation:** Jenny C. Censin, Reedik Mägi.

**Formal analysis:** Jenny C. Censin, Teresa Ferreira, Sara L. Pulit.

**Investigation:** Jenny C. Censin, Jonas Bovijn.

**Methodology:** Jenny C. Censin, Teresa Ferreira, Sara L. Pulit, Michael V. Holmes, Cecilia M. Lindgren.

**Project administration:** Jenny C. Censin, Michael V. Holmes, Cecilia M. Lindgren.

**Resources:** Cecilia M. Lindgren.

**Software:** Jenny C. Censin, Jonas Bovijn, Teresa Ferreira, Sara L. Pulit, Anubha Mahajan.

**Supervision:** Michael V. Holmes, Cecilia M. Lindgren.

**Validation:** Jenny C. Censin, Michael V. Holmes.

**Visualization:** Jenny C. Censin.

**Writing – original draft:** Jenny C. Censin.

**Writing – review & editing:** Jenny C. Censin, Sanne A. E. Peters, Jonas Bovijn, Teresa Ferreira, Sara L. Pulit, Reedik Mägi, Anubha Mahajan, Michael V. Holmes, Cecilia M. Lindgren.

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
