## [Decision Letter · Decision Letter 0]

8 Aug 2019

Dear Dr Lindgren,

Thank you very much for submitting your Research Article entitled 'Causal relationships between obesity and the leading causes of death in women and men' to PLOS Genetics. Your manuscript was fully evaluated at the editorial level and by independent peer reviewers. The reviewers appreciated the attention to an important topic but identified some aspects of the manuscript that should be improved.

We therefore ask you to modify the manuscript according to the review recommendations before we can consider your manuscript for acceptance. Your revisions should address the specific points made by each reviewer.

[LINK]

Yours sincerely,

Chris Cotsapas, PhD

Associate Editor

PLOS Genetics

Hua Tang

Section Editor: Natural Variation

PLOS Genetics

Reviewer's Responses to Questions

**Comments to the Authors:**

Reviewer #1: Overview: In this work, the authors employ genetic risk scores for obesity related traits (body mass index, waist-hip ratio, and BMI-adjusted waist-hip ratio) as instrumental variables in Mendelian Randomization analysis of leading causes of mortality in the UK biobank. The authors accounted for sex in construction of the genetic risk scores and considered potential mediators of mortality risk, e.g smoking and blood pressure traits.

The authors report that obesity causes an increase in risk for many leading causes of death, including coronary artery disease, stroke, chronic obstructive pulmonary disease, lung cancer, type 2 and 1 diabetes mellitus, non-alcoholic fatty liver disease, chronic liver disease and renal failure. For the risk of type 2 diabetes, chronic obstructive pulmonary disease and renal failure, authors find that the effects of obesity on risk differ between men and women.

Comments:

* Except for the number of variants, I could not find information about the exact genetic markers used in the construction of genetic risk scores. It would be useful to have a supplementary data sheet with the annotations of the variants, utilized in specific GRS’.

* The sex-specific approach to GRS construction is a potentially useful tool which is very poorly explored in the literature. It could be useful if authors included some information on the utility of the chosen sex-specific GRS-construction approach in and it's possible refinement and future applications in the Discussion section (e. g. under Strength and Weaknesses).

Reviewer #2: This manuscript addresses the important genetic question of whether the sex dimorphism in obesity-related traits causes a sex difference in risks for complex disorders. To answer this question, the authors apply the Mendelian randomization (MR) technique with sex-specific genetic instrument variables (IV) constructed using published GWAS summary statistics. While I find the sex dimorphism and its potential implication to pathophysiology can have a major implication to our understanding of the disease genetics, my enthusiasm is diminished by a major concern below.

I am concerned that the proposed method cannot answer how much of the observed sex difference in obesity-mediated disease risk is due to sex dimorphism vs social/environmental differences between sex. For instance, the authors report the sex difference for the risk of being a smoker depending on BMI and WHR. However, how much of this is due to the physiological difference and how much is due to gender norm and social pressure? If it is mostly the latter, the authors can still claim it is genetic - sex is genetically determined, - but my excitement on this work will become diminished. The authors attempt to address this issue in "potential mechanisms" section, but this issue seems to remain largely inconclusive. My concern is not narrowly focused on the issue of smoking status potentially explaining the sex difference in lung disease risks, which is discussed in the Discussion section. Rather, I am concerned that social confounder may affect both exposure (obesity traits) and disease risk and that sex is correlated with this confounder. And this possibility has to be reasonably ruled out in the analyses.

I am concerned that the CAD did not show strong sex differences. CAD is probably one of the most sex-dimorphic disorders, showing a striking sex difference in the prevalence by age. Is this perhaps because the covariate modeling of age, age^2 and sex not enough to model this prevalence pattern?

I would like to see if Winner's curse in obesity traits did not affect the sex-specific differences. They show that using unweighted GRS does not affect the conclusion, but the unweighted GRS could still be biased toward one sex because of p-value thresholding effect.

**Have all data underlying the figures and results presented in the manuscript been provided?**

Reviewer #1: Yes

Reviewer #2: Yes

PLOS authors have the option to publish the peer review history of their article (what does this mean?). If published, this will include your full peer review and any attached files.

Reviewer #1: No

Reviewer #2: No

---

## [Editor Report · Decision Letter 1]

9 Sep 2019

Dear Dr Lindgren,

We are pleased to inform you that your manuscript entitled "Causal relationships between obesity and the leading causes of death in women and men" has been editorially accepted for publication in PLOS Genetics. Congratulations!

Yours sincerely,

Chris Cotsapas, PhD

Associate Editor

PLOS Genetics

Hua Tang

Section Editor: Natural Variation

PLOS Genetics

Comments from the reviewers (if applicable):

**Data Deposition**

http://datadryad.org/submit?journalID=pgenetics&manu=PGENETICS-D-19-00903R1

**Press Queries**

---

## [Editor Report · Acceptance letter]

4 Oct 2019

PGENETICS-D-19-00903R1 

Causal relationships between obesity and the leading causes of death in women and men 

Dear Dr Lindgren, 

We are pleased to inform you that your manuscript entitled "Causal relationships between obesity and the leading causes of death in women and men" has been formally accepted for publication in PLOS Genetics! Your manuscript is now with our production department and you will be notified of the publication date in due course.

With kind regards,

Kaitlin Butler

PLOS Genetics

On behalf of:
